# Use of Food Spoilage and Safety Predictor for an “A Priori” Modeling of the Growth of Lactic Acid Bacteria in Fermented Smoked Fish Products

**DOI:** 10.3390/foods11070946

**Published:** 2022-03-25

**Authors:** Angela Racioppo, Daniela Campaniello, Milena Sinigaglia, Antonio Bevilacqua, Barbara Speranza, Maria Rosaria Corbo

**Affiliations:** Department of Agriculture, Food, Natural Resources and Engineering (DAFNE), University of Foggia, 71122 Foggia, Italy; angela.racioppo@unifg.it (A.R.); daniela.campaniello@unifg.it (D.C.); milena.sinigaglia@unifg.it (M.S.); antonio.bevilacqua@unifg.it (A.B.); mariarosaria.corbo@unifg.it (M.R.C.)

**Keywords:** predictive microbiology, FSSP, DoE, smoke, fermentation, fish

## Abstract

Fermentation is one of the oldest methods to assure the safety and quality of foods, and to prolong their shelf life. However, a successful fermentation relies on the correct kinetics depending on some factors (i.e., ingredients, preservatives, temperature, inoculum of starter cultures). Predictive microbiology is a precious tool in modern food safety and quality management; based on the product characteristics and the conditions occurring in food processing, the inactivation of or increase in microbial populations could be accurately predicted as a function of the relevant intrinsic or extrinsic variables. The main aim of this study was the optimization of the formula of a smoked fermented fish product using predictive modeling tools (tertiary and secondary models) in order to define the role of each factor involved in the formulation and assure a correct course of fermentation. Product optimization was conducted through the software Food Spoilage and Safety Predictor (FSSP), by modeling the growth of lactic acid bacteria (LAB) as a function of some key parameters such as temperature, pH, salt, liquid smoke, carbon dioxide, and nitrites. The variables were combined through a fractional design of experiments (DoE) (3^k-p^), and the outputs of the software, i.e., the maximal growth rate (μ_max_) and the time to attain the critical threshold (t_crit_), were modeled through a multiple regression procedure. The simulation, through FSSP and DoE, showed that liquid smoke is the most critical factor affecting fermentation, followed by temperature and salt. Concerning temperature, fermentation at 20–25 °C is advisable, although a low fermentation temperature is also possible. Other parameters are not significant.

## 1. Introduction

The application of fermentation to fish goes back many thousands of years with evidence of fermented fish products in ancient Greece (*aimeteon*) and in the Roman era (*garum*). However, although the fermentation process was once only used as a preservation method, today, it is also used for health purposes. In fact, a recent trend in the design of innovative fish products is the individuation of fish formulas fermented with appropriate starter cultures (sometimes with probiotic or other functional traits) to produce foods with improved sensory attributes, high nutritive value, and health benefits [1,2].

Fermented fish products usually have unique characteristics, especially in terms of aroma, flavor, and texture: this is the result of the transformation of organic materials into simpler compounds by the activity of microorganisms or enzymes during the fermentation process. The fermented products are more digestible, at the same time preserving their nutritional properties, even better than unfermented raw fish. These foods are a source not only of proteins, amino acids, and polypeptides, but also of minerals (calcium and iron), some B group vitamins, and, above all, polyunsaturated fatty acids [3].

Unfortunately, most fermented fish products are still local and not so easily found nationwide, with Asia and Africa being the main producers; only some fish sauces and shrimp pastes are widely known in Europe [4]. Each country has its own types of fermented fish products characterized by different production processes and formulations.

Recently, these products have been gaining increasing interest among consumers due to their healthy characteristics supported by several studies in the literature [3,5,6,7,8,9,10,11,12,13,14,15]. Apart from the well-known health effects of fish meat, correct fermentation, led by pro-technological microorganisms, could result in the biofortification of amino acids and peptides with some physiological and beneficial functions (antioxidant, antihypertensive, antiproliferative, hypoglycemic, immune-stimulating, and anticoagulant effects) [1]. In addition, a correct and predictable process piloted by starter cultures, either autochthonous or commercial, can limit the production of undesired metabolites and produce a good product; the choice to rely the quality of fermented products on spontaneous fermentation (natural microbiota) is no longer advisable, because of the very high risk of incurring arrested fermentation and health problems. This is why recent research is increasingly focused on the selection of microorganisms able to guarantee the best performances [16,17,18]. The most promising microorganisms selected as starters are generally those that are isolated from the native microbiota of traditional products since they are well adapted to the environmental conditions of the considered food [19,20,21].

Regardless of the type of starter, the correct course of fermentation is essential to produce a safe and high-quality product. It is also crucial to define the effects of each factor involved in the formulation (ingredients, preservatives, temperature, inoculum of starter cultures) on the process of evolution. In this context, predictive microbiology stands up as a precious tool in modern food safety and quality management. Based on the product characteristics and the conditions occurring in food processing, the inactivation of or increase in microbial populations in foods, as a function of the relevant intrinsic or extrinsic variables, could be accurately predicted [22]. A new frontier goal in predictive microbiology is the use of models and databases for product optimization, as also shown by the authors of some seafood applications [23,24].

Generally, the optimization of new formulas relies on different steps and phases and could start with an “a priori” modeling as a step prior to actual challenge tests at the laboratory level, followed by scaling up and optimization. However, before planning and executing some experiments, it is advisable to understand the role of most variables involved, and to perform a screening to exclude less significant or non-significant factors. The reduction in the number of variables is important from a mathematical point of view, because the experience of authors suggests that the use of many variables could lead to confounding phenomena; in addition, a priori modeling could help researchers minimize the number of experiments required to be performed in the lab, as well as helping them to better define the conditions of assays.

Therefore, the main aim of this study was an “a priori” optimization of the formula of a smoked fermented fish product using predictive modeling tools to define the role of each factor involved in the formulation (temperature, pH, salt, liquid smoke, carbon dioxide, and nitrites) for a correct course of fermentation. The variables were combined through a fractional design of experiments (DoE) (3^k-p^), and the outputs of the software, i.e., the maximal growth rate (μ_max_) and the time to attain the critical threshold (t_crit_), were modeled through a multiple regression procedure.

## 2. Materials and Methods

### 2.1. Software and Design

Product optimization was conducted through Food Spoilage and Safety Predictor (FSSP), a free software package available as an additional tool in ComBase or on the website of the developers (http://fssp.food.dtu.dk/, accessed on 6 September 2021).

The focus was the modeling of lactic acid bacteria (LAB) growth as a function of some key parameters for a fermented fish product including intrinsic factors or extrinsic factors (temperature and pH), processing parameters (salt, liquid smoke, nitrites, or carbon dioxide), and parameters connected to LAB growth (relative lag time). The choice of liquid smoke relies on the fact that the consumer appeal towards smoked fish products is increasing and the use of liquid smoke instead of the traditional method is advisable because it is more eco-friendly, while carbon dioxide is crucial to simulate fermentation and storage under anoxic conditions. Finally, nitrites are the preservatives generally used in fermented meat products to counteract *Clostridium botulinum* growth. The variables were combined through a fractional design (3^k-p^), and each factor had three levels (minimum or “−1”, mean value or “0”, and maximum or “+1”), as shown in Table 1. The combination of these variables resulted in a design consisting of 243 runs, as shown in the Appendix A. The input conditions for modeling were as follows: initial concentration of LAB, 5 log CFU/g; maximal population density, 9 log CFU/g; critical threshold, 8.9 log CFU/g; weak acids at 0 ppm. For modeling purposes, weak acids were excluded to reduce the number of variables to 7; in a mixed design, the randomization of 7 variables set at 3 levels produces a total of 243 combinations and makes the estimation of both the individual effects of each variable and their two-way interactions possible.

An increase in the number of variables causes an increase in the number of the combinations of the design (up to 729 combinations with 9 variables); a secondary effect of increasing the number of variables is the possibility of a statistical artifact due to the high number of individual and interactive terms. Thus, weak acids were not used for this design, because they are added at the end, and not throughout fermentation, to prolong the shelf life of the product.

The tool works on psychrotolerant LAB (*Latilactobacillus sakei*; *Latilactobacillus curvatus*; *Lactobacillus* spp.; and other related genera from autochthonous microbiota of fish) [25].

### 2.2. Modeling

All 243 combinations built by the randomization of the variables were used as inputs for the software, and for each combination, the maximal growth rate (μ_max_) and the time to attain the critical threshold (t_crit_), as an indirect measure of the time required to attain the steady state, were evaluated; the values are presented in Appendix A. Then, μ_max_ and t_crit_ were modeled through a multiple regression procedure, using the option DoE (design of experiments) of the software Statistica for Windows (Version 7, Tulsa, OK, USA), to obtain a second-order model as follows:(1)$$y=B_{0}+\sum B_{i}x_{i}+\sum B_{ii}x_{i}{}^{2}+\sum B_{ij}x_{i}x_{j}$$
where “*y*” is the modeled dependent variable (μ_max_ and t_crit_); *B_i_*, *B_i_*_i_, and *B_ij_* are the coefficients of the model, associated with independent variables; the term “*x_i_*” highlights the individual effect of the predictor; the symbols “*x_i_*^2^” and “*x_i_x*_j_” indicate the quadratic and interactive effects. Variables showing a significance <95% (*p* > 0.05) were not included in the equation by the software; moreover, due to the high number of variables, interactive terms were excluded.

The effect of each independent variable on the outputs was also evaluated through the individual desirability functions, estimated as follows:(2)$$d=\left(\begin{array}{cc} 0, & y\leq y_{unfav}\\ \frac{y-y_{unfav}}{y_{fav}-y_{unfav}} & \;y_{unfav}\leq y\leq y_{fav}\\ 1, & y\geq y_{fav} \end{array}\right)$$
where *y_unfav_* and *y_fav_* are the most unfavorable (high value of t_crit_ and low value of μ_max_) and the most favorable (low value of t_crit_ and high value of μ_max_) values of the dependent variables.

## 3. Results and Discussion

The use of tertiary models to predict safety and food shelf life has also been regarded by the European Union as a tool for quality assurance. The tertiary model used in this work was originally designed for seafood products [26,27], but it has been recently updated for other foods (for example, cheese), and the name has been changed to FSSP. In the latest version, it contains a collection of several options/models, including RRS models (relative rate of spoilage), and models to predict spoilers (*Photobacterium phosphoreum*, *Morganella psychrotolerans*, *Morganella morganii*, *Shewanella putrefaciens*) and biogenic amine formation, pathogens (*Listeria monocytogenes*), and lactic acid bacteria. As observed in the study of Mejlholm et al. [28], predictions performed through the FSSP software have good precision when the complexity of the growth model matches the complexity of the foods of interest; this was also confirmed by the study of Bolívar et al. [29] during the modeling of the growth of *L. monocytogenes* in Mediterranean fish species, where the software was used to estimate μ_max_ based on the values of pH, a_w_, storage temperature, and atmospheric conditions of the studied fish species. The goodness of fit of FSSP was regarded as acceptable for *Lactobacillus* spp. (now *Lactobacillus* and related genera); in addition, the tool is flexible and focuses on a high number of parameters, and for some of them (weak acids and preservatives), it is reliable a substitution of some compounds with other ones chosen by users [25].

LAB, mainly psychrotolerant strains, grow at high concentrations during the storage of seafood, resulting in the production of off-odors, gas, slime, and undesired metabolites [25,30]. However, there are various positive and pro-technological species and strains able to contribute to shelf life and healthy characteristics, because of their metabolism and health properties [21,23,30,31]. Apart from strain characterization and validation in some matrices, to the best of the authors’ knowledge, there are no models able to predict the performances of LAB during fish fermentation. These models, hereby labeled as a priori functions, could be extremely useful in the first phase of product design, because they could help researchers to choose only some variables for laboratory confirmation.

It is important to include in such models both the matrix’s variables and processing factors (pH, temperature, carbon dioxide), including preservatives used in the process (salt and nitrites). Another variable hereby tested was the relative lag time (RLT). This parameter is different from the classical lag time and relies on the physiological state of cells introduced into a new environment (for a starter culture from a bioreactor to food) and should be read as the “amount of work done by cells to adapt to the new environment” [32]. From a theoretical and practical point of view, RLT is at its minimum when the microorganism has a very low lag phase and grows at its optimal growth rate, and could be 0, when the microorganism has no lag phase.

The inclusion of RLT as a main variable of the model relies on the fact that, in a guided fermentation, it is crucial to inoculate well-adapted microorganisms in the matrix (low RLT), or to use them in an environment with the same characteristics of the media used to produce their biomass. This factor was described in the past as the physiological function of the Baranyi and Roberts model and was shown to strongly affect the growth curve [33].

The steps taken in this research are similar to those used when a second-level model of predictive microbiology is developed: the first step was the evaluation of the growth curves in some conditions; then, the fitting parameters of the growth curves were modeled as a function of pH, temperature, composition, etc. In this research, FSSP substitutes for the first step.

Figure 1 shows an example of the growth curves (combinations S34 and S41) predicted for LAB using FSSP. For each combination studied (S1), the outputs of the software (μ_max,_ maximal growth rate, and t_crit,_ time to attain the critical threshold of 8.9 log CFU/g) were recorded and subsequently used as input data for a multiple regression procedure.

The growth rate is the classical parameter for secondary models, but t_crit_ was added in this research. To the authors’ knowledge, this is the first time t_crit_ has been added in a modeling study of LAB cultures. In a simple way, t_crit_ could be defined as the time to attain the steady state; thus, it is an indirect measure of the performances of starter cultures and of their acidification kinetics. Moreover, the time to attain the steady state could have other implications related to the Jameson effect: a dominant group (a starter culture) could stop the growth of the other subpopulations (spoiling microorganisms) and induce a stationary phase, thus reducing the maximum level they could attain [34]. For a starter culture, the Jameson effect, and thus the time to attain the steady state, could have a strong effect on the inhibition of undesirable microorganisms and on bioprotective effects. 

The first output of the statistical approach used was the table of standardized effects (Table 2), which shows the statistical weight of each studied factor; in particular, it summarizes the effects of the linear (L), quadratic (Q), and interactive terms of RLT (relative lag time), temperature, concentration of NaCl, liquid smoke, and the amount of nitrite and CO_2_ at equilibrium in the headspace on the maximal growth rate (μ_max_) and on t_crit_ of LAB in fish products. The standardized effect was evaluated as the ratio of the mathematical coefficient of each factor (from multiple regression) vs. its standard error.

For the maximal growth rate, the most significant factor was liquid smoke, followed by temperature and salt; pH, nitrites, RLT, and CO_2_ did not play a significant role. For the time to attain the critical threshold (t_crit_), the data point out the significance of liquid smoke, temperature, RLT, and salt. Although only for qualitative purposes, the main benefit of this model, compared to the boundary approach used for LAB in FSSP [25], is that it does not offer a negative overview (combinations of parameters causing growth inhibition); the ratio behind this approach is a positive method (from a mathematical point of view), that is, some of the factors and interactions are related to growth and are the parameters able to cause an increase in the dependent variables (rate and t_crit_).

A quantitative estimation of the role of the significant variables could be gained through the use of surface response plots. Concerning the interactions of liquid smoke × temperature (Figure 2a) and liquid smoke × salt (Figure 2b), the growth rate was at its maximum at the highest value of temperature (coded level “+1”, 25 °C) and with the lowest value of salt and liquid smoke; however, the model predicted the complete inhibition of LAB growth only for a few combinations of temperature and smoke (level “−1” for temperature and “0.8–1” for liquid smoke, corresponding to actual values of 10 °C and 35–40 ppm of liquid smoke). The surface response plot for the interaction of nitrite × liquid smoke (Figure 2c) underlines the non-significant effects of nitrites. The effect of liquid smoke on the growth of LAB was also confirmed by the results obtained for t_crit_, as the highest values of this parameter were predicted for amounts of liquid smoke over 20–25 ppm (data not shown).

Surface plots are a good tool for treating data from a multiple regression, and for highlighting the individual and quadratic terms of each factor, as well as for showing the interactive effects amongst the different independent variables. However, despite these benefits, it is not possible to see the effect of each variable excluding the others, i.e., the effect of each factor is linked to the effects of other variables. Thus, to this scope, other approaches could be used, such as desirability which is a dimensionless parameter able to provide an answer to the following issue: how much is an output desired? [35] In addition, through mathematical extrapolation, desirability is an important tool to highlight critical values for each factor.

Figure 3 and Figure 4 show the desirability profile for μ_max_ and t_crit_, respectively; the profile highlights the different quantitative weights of each studied variable. Focusing on the results obtained for μ_max_ (Figure 3), only temperature, salt, and liquid smoke are able to affect fermentation, whereas other parameters such as pH, use of anoxic conditions, and nitrites are not significant. Concerning temperature, fermentation at 20–25 °C is advisable, although a low fermentation temperature is also possible (10–15 °C). This result is corroborated in numerous other studies using mathematical models to predict the growth of different microorganisms in foods at various storage temperatures [36,37,38], where a strong dependency was found between microbial growth and the storage temperature assayed.

Although a temperature of 25 °C is generally not the optimal temperature of LAB, and a further increase in desirability for higher temperatures could be hypothesized, the choice of the range was due to two main reasons: (i) higher temperatures could have masked the effect of other variables because of the strongest effect of temperatures for starter culture kinetics [39]; (ii) fish fermentation is advisable at a lower temperature to avoid massive production of undesirable compounds, such as biogenic amines [40]. Finally, 25 °C is the maximum level of temperature for LAB in FSSP.

As expected, the effect of salt was less significant than temperature (at least in the range tested in this study, 0–6%): lactic acid bacteria, in fact, are able to tolerate high salt concentrations, and this tolerance gives them an advantage over other less tolerant species, allowing a rapid start of fermentation [21]. The results (see box of NaCl in Figure 3) show that the decrease in μ_max_ begins to be significant beyond a salt concentration of 3%; however, 3–3.5% (*w*/*w*) is the salt concentration recommended by the Codex Alimentarius [41] in smoked-flavored fish products to avoid the proliferation of spoilage bacteria. Both Figure 3 and Figure 4 point out that a smoke concentration >20 ppm (coded level 0) is not advisable because it could result in a significant delay of starter growth. This inhibitory effect is probably attributable to the bactericidal effects of smoke components (phenols, polycarboxylic acids) on most types of bacteria and/or fungi, including LAB [42]. Many other recent studies were focused on the use of liquid smoke on fish products [43,44,45], since consumer appeal towards these foods has been increasing recently. The traditional smoking method is under investigation since it can cause the production of harmful compounds, such as polycyclic aromatic hydrocarbons, mainly benzo(α)pyrene [46]. Consequently, the use of liquid smoke is advisable because it is more eco-friendly, requires a shorter smoking time, and ensures a longer product durability [44].

In this study, model building was based on a completely randomized design, which was able to estimate possible interactive and quadratic effects. This type of design is useful for screening purposes, and to reduce the number of variables; however, for robust predictive equations, other designs (for example, central composite design) are advisable, because a higher number of levels allows for good precision in the prediction, at least in the ranges tested. Thus, further investigations are required in this field, in order to assess the significance of other factors not considered in FSSP (food structure, food components, effects of natural microbiota, etc.) which could strongly and significantly affect the growth/survival of lactic acid bacteria.

## 4. Conclusions

The design of a fermented product is a complex process, and the definition of the formulation is a critical step above all for the performances of the starter cultures. The use of LAB is a promising way to valorize seafood products and to design safe, stable, and consumer-appealing products; however, some variables should be considered.

The use of liquid smoke and the amount of phenols are critical, because the simulation through FSSP suggested that there is a critical amount (20 ppm) after which the fermentation could experience a delay.

Concerning temperature, fermentation at 20–25 °C is advisable because of higher growth rate values, although a low fermentation temperature is also possible. The model, in fact, also predicted growth at 10–15 °C.

RLT could strongly affect the performances of a starter culture, thus suggesting that the physiological state of cells could affect their performances during fish fermentation.

Other parameters are not significant (at least those tested in this case study: pH, use of anoxic conditions, nitrites), and if they are included in the formulation, their amount should be determined through other considerations (chemistry, costs, safety for consumers, etc.).

This model could be the background for future studies devoted to the optimization of fish fermentation, as it offers some details on the effect of many factors, although it is important to consider that each a priori modeling should be followed by validation, to check if the mathematical effects have the same weight in real systems, or if other factors could play a role.

## Figures and Tables

**Figure 1 foods-11-00946-f001:**
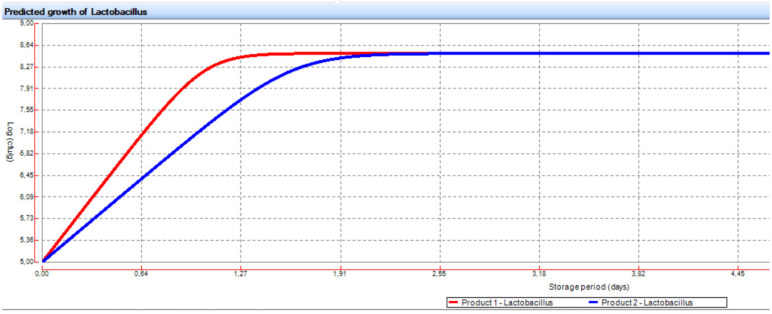
Example of growth curves (combinations S34 and S41) predicted for LAB using FSSP.

**Figure 2 foods-11-00946-f002:**
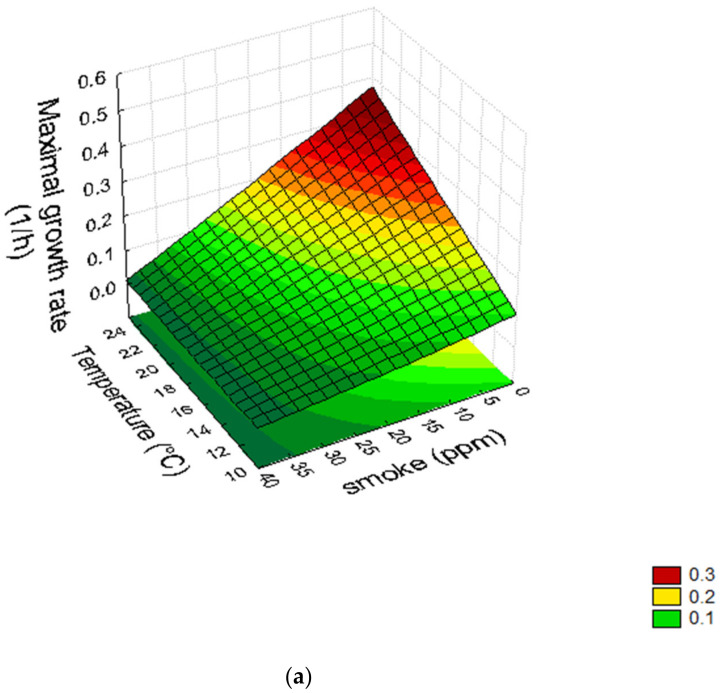
Surface response plots for the interactions liquid smoke × temperature (**a**), liquid smoke × salt (**b**), and nitrite × liquid smoke (**c**).

**Figure 3 foods-11-00946-f003:**
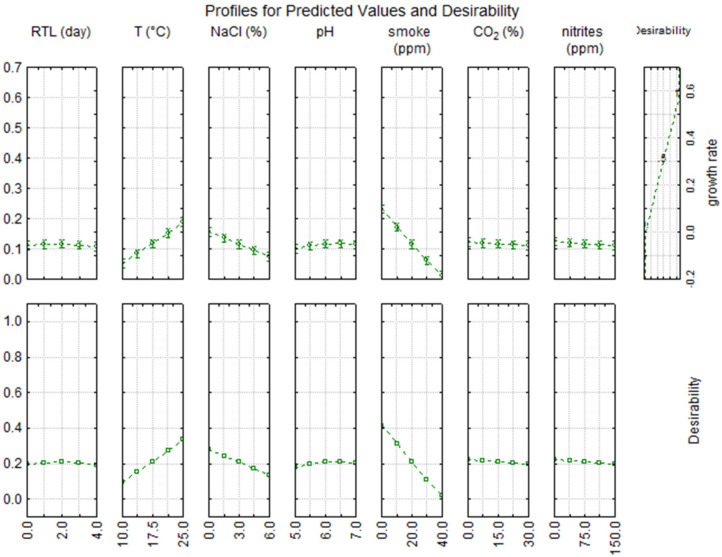
Desirability profile for the maximal growth rate (μ_max_) of lactic acid bacteria in fish products.

**Figure 4 foods-11-00946-f004:**
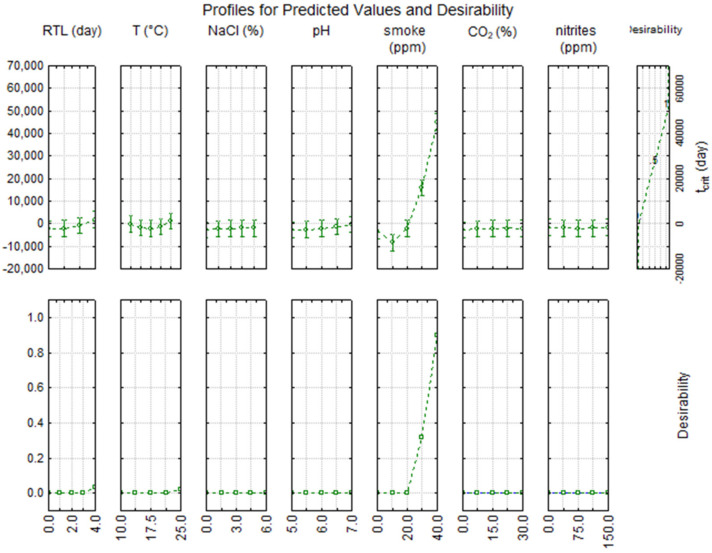
Desirability profile for the time to attain the steady state (t_crit_) of lactic acid bacteria in fish products.

**Table 1 foods-11-00946-t001:** Coded levels of the variables.

	−1	0	+1
RTL (days)	0	2	4
Temperature (°C)	10	17.5	25
NaCl (%)	0	3	6
pH	5	6	7
Smoke (ppm)	0	20	40
CO_2_ (%)	0	15	30
Nitrite (ppm)	0	75	150

**Table 2 foods-11-00946-t002:** Standardized effects of linear (L), quadratic (Q), and interactive terms of RLT (relative lag time), temperature, concentration of NaCl, liquid smoke, and the amount of nitrite and CO_2_ at equilibrium in the headspace on the maximal growth rate (μ_max_) and time to attain the steady state (t_crit_) of lactic acid bacteria in fish products. The standardized effect was evaluated as the ratio of the mathematical coefficient of each factor (from multiple regression) vs. its standard error. The degrees of freedom for the *t*-test were 204; R^2^_ad_, regression coefficient corrected for multiple regression. * Not significant.

	μ_max_	t_crit_
(1) RTL (L)	- *	5.030
RTL (Q)	-	-
(2) T (L)	29.729	−7.516
T (Q)	-	−3.191
(3) NaCl (L)	−17.806	-
NaCl (Q)	-	-
(4) pH (L)	3.109	-
pH (Q)	2.455	-
(5) Smoke (L)	−48.844	8.146
smoke (Q)	-	−5.215
(6) CO_2_ (L)	−2.502	-
CO_2_ (Q)	-	-
(7) nit (L)	−2.943	-
nit (Q)	-	-
1L by 2L	-	−4.980
1L by 3L	-	-
1L by 4L	-	-
1L by 5L	-	5.023
1L by 6L	-	-
1L by 7L	-	-
2L by 3L	−10.036	-
2L by 4L	-	-
2L by 5L	−24.746	−8.288
2L by 6L	-	-
2L by 7L	-	-
3L by 4L	−2.843	-
3L by 5L	14.087	-
	μ_max_	t_crit_
3L by 6L	-	-
3L by 7L	-	-
4L by 5L	−4.546	-
4L by 6L	-	-
4L by 7L	-	-
5L by 6L	-	-
5L by 7L	2.378	-
6L by 7L	-	-
R^2^_ad_	0.947	0.551

## Data Availability

Not applicable.

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
