# Peer review of "Use of Food Spoilage and Safety Predictor for an “A Priori” Modeling of the Growth of Lactic Acid Bacteria in Fermented Smoked Fish Products"

_foods, 2022, doi:10.3390/foods11070946_

Round 1

Reviewer 1 Report

Was the model only hypothetic or any fermentations were made to build the hypostasis? It is not clear how many and which of the 243 runs were performed to obtain the data to build the model. This information is very important. 
Which LAB(s) were used as starter culture?
Rewrite the sentences L.12-14 in the abstract or L.66-69 in the introduction,  because both sentences like very similar.

Author Response

Was the model only hypothetic or any fermentations were made to build the hypostasis? It is not clear how many and which of the 243 runs were performed to obtain the data to build the model. This information is very important. 

Replied (l. 130-133)

Which LAB(s) were used as starter culture?

Replied (l. 125-126)

Rewrite the sentences L.12-14 in the abstract or L.66-69 in the introduction, because both sentences like very similar.

The sentence in the abstract was modified (l. 12-13)

Reviewer 2 Report

Peer Review report 1 on: “Use of Food Spoilage and Safety Predictor to Assess the Growth of Starter Lactic Acid Bacteria in a Fermented Smoked Fish Product”.

  1. Original submission.
    • Recommendation

Minor revisions

  1. Comments to Author

Manuscript Number: Foods-1639397

Title: Use of Food Spoilage and Safety Predictor to Assess the Growth of Starter Lactic Acid Bacteria in a Fermented Smoked Fish Product

Overview and general recommendations:

The manuscript describes the use of the software FSSP to develop a formula for smoked fermented fish, with the assistance of optimization tools to predict the microbial populations in the product and to assure an appropriate fermentation process for a safe and high-quality product. The document describes in detail the necessity of tools for predicting the processes in foods before the actual manufacturing and the existence of tools like FSSP, offering even evidence of other studies based on those tools. The manuscript is written with clarity and describes properly the background and the objectives of the study. As well, the document is written in very good English which makes it easy to understand.

My only concerns are two. 1) Was there any biological/practical validation of the results? Are the results in a theoretical level?  2) The conclusion does not completely match with the objectives, declaring even some points by first time in the whole document, which I believe had to be discussed before.    

The section of results and discussion may be improved. Since there is not apparently a biological validation of the results, the discussion may include and analyze some biological studies supporting the results of the proper study, to make stronger statements on the mathematical model.    

Author Response

The manuscript describes the use of the software FSSP to develop a formula for smoked fermented fish, with the assistance of optimization tools to predict the microbial populations in the product and to assure an appropriate fermentation process for a safe and high-quality product. The document describes in detail the necessity of tools for predicting the processes in foods before the actual manufacturing and the existence of tools like FSSP, offering even evidence of other studies based on those tools. The manuscript is written with clarity and describes properly the background and the objectives of the study. As well, the document is written in very good English which makes it easy to understand.

My only concerns are two. 1) Was there any biological/practical validation of the results? Are the results in a theoretical level? 

The research was an a priori modelling, as better stated in the introduction; the reasons beyond this research were reported (l. 76-84; l. 184-188)

2) The conclusion does not completely match with the objectives, declaring even some points by first time in the whole document, which I believe had to be discussed before.   

Slight correction to make this section adherent to Discussion 

The section of results and discussion may be improved. Since there is not apparently a biological validation of the results, the discussion may include and analyze some biological studies supporting the results of the proper study, to make stronger statements on the mathematical model.  

Discussion was improved  

Reviewer 3 Report

The manuscript "Use of Food Spoilage and Safety Predictor to Assess the Growth of Starter Lactic Acid Bacteria in a Fermented Smoked Fish Product" by Angela et al.

What's the novelty of the work. It should be stated clearly and is the main drawback of the manuscript to point out the important point of the study. Why was it undertaken?

Why not organic acids as one of the variables included in the study?

Line 88-94: This sentence should be divided as it is way too long.

The highest temperature used in the study was 25C...And was suggested to have a maximum growth rate (Line 187). So what happens if the temperature is set beyond 25. I would suggest changing the temperature range if the best growth is coming at the highest temperature selected.

Table 1 is a repetition from supplementary excel sheet S1.

Author Response

The manuscript "Use of Food Spoilage and Safety Predictor to Assess the Growth of Starter Lactic Acid Bacteria in a Fermented Smoked Fish Product" by Angela et al.

What's the novelty of the work. It should be stated clearly and is the main drawback of the manuscript to point out the important point of the study. Why was it undertaken?

Replied (l. 76-84; l. 184-188)

Why not organic acids as one of the variables included in the study?

Replied (l. 115-123)

Line 88-94: This sentence should be divided as it is way too long.

Revised (l. 101-108)

The highest temperature used in the study was 25C...And was suggested to have a maximum growth rate (Line 187). So what happens if the temperature is set beyond 25. I would suggest changing the temperature range if the best growth is coming at the highest temperature selected.

We have replied to this issue (l. 288-294)

Table 1 is a repetition from supplementary excel sheet S1.

The coded levels were removed from supplementary materials and retained only in table 1

Reviewer 4 Report

Manuscript is clearly written and is easy to understand. In L56 - positive effects listed are to the best of my knowledge attributed also to 'normal' fish meat - so what is really added value of fermented fish meat?

In my opinion, supporting the results of the modeling experimentally would make a significant contribution to the importance of the study.

Author Response

Manuscript is clearly written and is easy to understand. In L56 - positive effects listed are to the best of my knowledge attributed also to 'normal' fish meat - so what is really added value of fermented fish meat?

The sentence was re-written (l. 53-56)

In my opinion, supporting the results of the modeling experimentally would make a significant contribution to the importance of the study.

The manuscript was improved by underlying a priori modelling was proposed and which is the benefit of this approach: see for example l. 76-84; l. 184-188

Round 2

Reviewer 3 Report

The manuscript has been substantially improved by careful revision of the authors. 

Hence, I recommend the article to be accepted in the present form.